# Application of Novel Transcription Factor Machine Learning Model and Targeted Drug Combination Therapy Strategy in Triple Negative Breast Cancer

**DOI:** 10.3390/ijms241713497

**Published:** 2023-08-31

**Authors:** Jianyu Pang, Huimin Li, Xiaoling Zhang, Zhengwei Luo, Yongzhi Chen, Haijie Zhao, Handong Lv, Hongan Zheng, Zhiqian Fu, Wenru Tang, Miaomiao Sheng

**Affiliations:** Laboratory of Molecular Genetics of Aging & Tumor, Medicine School, Kunming University of Science and Technology, Kunming 650500, China; pjy2021@stu.kust.edu.cn (J.P.); lhuimin@stu.kust.edu.cn (H.L.);

**Keywords:** triple-negative breast cancer, transcription factors, machine learning, prognosis, single-cell RNA-seq, combined targeted therapy

## Abstract

Transcription factors (TFs) have been shown to play a key role in the occurrence and development of tumors, including triple-negative breast cancer (TNBC), with a worse prognosis. Machine learning is widely used for establishing prediction models and screening key tumor drivers. Current studies lack TF integration in TNBC, so targeted research on TF prognostic models and targeted drugs is beneficial to improve clinical translational application. The purpose of this study was to use the Least Absolute Shrinkage and Selection Operator to build a prognostic TFs model after cohort normalization based on housekeeping gene expression levels. Potential targeted drugs were then screened on the basis of molecular docking, and a multi-drug combination strategy was used for both in vivo and in vitro experimental studies. The machine learning model of TFs built by *E2F8*, *FOXM1*, and *MYBL2* has broad applicability, with an AUC value of up to 0.877 at one year. As a high-risk clinical factor, its abnormal disorder may lead to upregulation of the activity of pathways related to cell proliferation. This model can also be used to predict the adverse effects of immunotherapy in patients with TNBC. Molecular docking was used to screen three drugs that target TFs: Trichostatin A (TSA), Doxorubicin (DOX), and Calcitriol. In vitro and in vivo experiments showed that TSA + DOX was able to effectively reduce DOX dosage, and TSA + DOX + Calcitriol may be able to effectively reduce the toxic side effects of DOX on the heart. In conclusion, the machine learning model based on three TFs provides new biomarkers for clinical and prognostic diagnosis of TNBC, and the combination targeted drug strategy offers a novel research perspective for TNBC treatment.

## 1. Introduction

Triple-negative breast cancer (TNBC) is a highly invasive and heterogeneous subtype of breast cancer that accounts for approximately 10–20% of breast cancer [1,2]. TNBC is characterized primarily by its adverse effects on the estrogen receptor (ER), progesterone receptor (PR), and human epidermal growth factor receptor 2 (HER2) [3]. Breast cancer is predisposed to local recurrence and distant metastases, and the prognosis of all subtypes of breast cancer is the worst [4,5]. Transcription factors (TFs) control chromatin and transcription through the recognition of specific DNA sequences to form complex systems that drive genome expression [6]. It controls cellular function, the body’s response to diseases, and the health of the body through the regulation of genes. Given the critical role of TFs, approximately 16% of almost 1600 TFs are implicated in a variety of cancers, including breast cancer [7,8]. TNBC patients have a poor response to therapy due to the absence of ER, PR, and HER2. Thus, there is an urgent need for personalized and reliable diagnosis and treatment methods [9,10,11]. The potential for TFs to serve as clinical biomarkers and their diagnostic and prognostic value has been previously reported [12]. To date, machine learning has been extensively used in TNBC; both random forest machine models for the prediction of immune checkpoint blockade [13] and the non-invasive detection of fluid by DNA methylation biomarkers [14] are feasible and efficient. However, there is still a lack of machine learning prediction models integrating TFs, so it will be exciting to build efficient and applicable TF machine learning prediction models. Given the complex regulatory properties of TFs, building a TF prediction model based on the TNBC gene will be more sensitive for predicting patient prognosis. Further research into targeted drugs based on ensemble models of TFs built by machine learning and the use of combined targeted drugs will contribute to research on treatment guidance for patients with TNBC. A detailed investigation of the mechanism of downstream TF regulation will also be useful in clarifying the prognostic mechanism predicted by this machine model and extending further TF research in the future.

In this study, bulk RNA-seq and single-cell RNA-seq (scRNA-seq) technologies were combined to conduct an exhaustive study of TFs in the TNBC cohort, and the specific research workflow is shown in Appendix A. Firstly, we conducted a differential analysis and weighted correlation network analysis (WGCNA) based on The Cancer Genome Atlas (TCGA) cohort of breast cancer patients (BRCA), and nine Hub TFs genes strongly related to TNBC were screened. In order to improve the applicability of the TFs prediction model, we used four housekeeping genes (*ACTB*, *GAPDH*, *TFRC*, *TUBB*) to normalize the expression levels of nine Hub TFs genes across the training and validation groups. The Least Absolute Shrinkage and Selection Operator (LASSO) regression algorithm was used to build the Hub TF feature score system (HTFSS), consisting of *E2F8*, *FOXM1*, and *MYBL2* genes. The sensitivity of HTFSS for prediction of prognosis was high, with 1-year Area Under Curve (AUC) values as high as 0.877 and 0.987 in the training and validation cohorts, respectively. Our assessment of multiple clinical features demonstrated that HTFSS can be used as an independent clinical risk factor for predicting worse patient prognosis and poor outcomes with immunotherapies. To extend the clinical application of HTFSS, a Nomogram was also constructed, and an online version of the interactive webpage was deployed to aid clinicians in their diagnosis. Based on scRNA-seq cohorts, we investigated the relationship between HTFSS and the tumor immune microenvironment (TIME). We found that *E2F8* and *FOXM1* were significantly associated with MKI67+ progenitor cells, while *MYBL2* was significantly associated with dendritic cells of the lymphoid lineage. Multiple TF target gene databases were used to explore the mechanism of regulation of the downstream target genes, and changes in pathway activity caused by imbalances in the three TFs were explored. Interestingly, when the three TFs were dysregulated, cell proliferation-related pathways, such as the PI3K-AKT and FOXO signaling pathways, were upregulated. Next, by combining the Comparative Toxicogenomics Database (CTD) and molecular docking, three targeted drugs (Doxorubicin (DOX), Trichostatin A (TSA), and Calcitriol) with strong binding activity to the expression proteins of the Hub TF gene were tested. We next tested the efficacy of the combination treatment in the murine model and at the cellular level. At the cellular level, RT-qPCR results showed that all three TFs were significantly upregulated in TNBC cell lines, and both DOX and TSA were able to effectively reduce their abnormal expression. DOX and TSA can also inhibit tumor cell progression by inhibiting the expression of all three TFs and regulating the PI3K-AKT and FOXO signaling pathways. In vivo and in vitro experimental results demonstrated that the combination of TSA + DOX was capable of effectively reducing the DOX dosage, which was beneficial to patients because it has been reported that DOX has multiple toxic side effects. TSA + DOX + Calcitriol can be used in combination to effectively reduce the toxic side effects of DOX on the heart, which is further optimization of the DOX, reducing the toxic side effects of the drug based on enhancing the antitumor effect. In conclusion, we believe that the clinical application of HTFSS provides a more sensitive and widely applicable biomarker of TFs integrated for TNBC, and the strategy of combined use of targeted drugs has the potential to optimize the antitumor effect of a single drug to some extent.

## 2. Results

### 2.1. Identification of TF Gene Sets Highly Associated with TNBC

First, 3396 differentially expressed genes (DEGs) were obtained from TCGA-BRCA, of which 1975 were upregulated and 1394 were downregulated (Figure 1A). WGCNA categorized highly related DEGs into nine modules with different colors (Figure 1B). Pearson correlation analysis was then carried out between the PAM50 classification phenotype of the BRCA modules and nine WGCNA modules (Figure 1C). Black, green, blue, and red were found to have a significant positive correlation with the Basal type phenotype, and the green correlation was the highest at 0.89. We then performed an additional Pearson correlation analysis between DEGs in the above four modules, and the Basal-like phenotype as well as 142 Hub genes, were ultimately confirmed (Figure 1D). The Kyoto Encyclopedia of Genes and Genomes (KEGG) pathway enrichment results showed that the Hub gene set was highly enriched in the cell cycle, the p53 signaling pathway, the microRNA in cancer, and other pathways related to cancer (Appendix A). Likewise, in the Gene Ontology (GO) enrichment results for the Hub gene, terms related to cell division, such as organelle division, chromosome separation, and tubulin binding, are highly enriched (Appendix A), suggesting that the Hub gene is tightly linked to tumor development. Lastly, nine Hub TF genes were identified by Venn diagramming (Figure 2A), which indicated that they played an essential role in the regulation of tumor development.

### 2.2. HTFSS Has Excellent Prognostic Prediction Performance

With the goal of further identifying Hub TF genes that may significantly influence the prognosis of TNBC patients, we used the LASSO regression algorithm to reduce the size of the nine Hub TF genes and performed a 10-fold cross-validation (Figure 2B,C). Prior to data analysis, cohort data were normalized by the expression of four housekeeping genes to enhance the applicability of HTFSS. The next step was to confirm the existence of the system consisting of three Hub TF genes: HTFSS = 0.046 × *MYBL2*_Exp_ − 0.010 × *FOXM1*_Exp_ − 0.295 × *E2F8*_Exp_. Next, the predictive value of the HTFSS was evaluated by the time-dependent Receiver Operating Characteristic (ROC) curve. The results demonstrated that the AUC value for the patient’s first-year survival rate was as high as 0.877, and the AUC values for survival at 3, 5, and 7 years were 0.699, 0.587, and 0.663, respectively (Figure 2D). The results described above were also confirmed in the validation set, and the largest predicted AUC value in the first year was 0.987, and the AUC values in the third, fifth, and seventh years for survival were 0.546, 0.637, and 0.614, respectively (Figure 2F). After grouping patients based on the HTFSS cohort median, Kaplan–Meier survival curves revealed that the High-Score group’s survival rate was significantly lower than that of the Low-Score group over the same period of time (Figure 2E,G). The HTFSS and patient death distributions also explained the poorer prognosis of High-Score patients, even though the expression heatmap suggested that the poor prognosis was driven by the upregulation of three Hub TF genes (Figure 2H,I). This study suggests that HTFSS, composed of the three upregulated Hub TF genes, has excellent prognostic and predictive value and can accurately predict a poor prognosis for patients.

### 2.3. Application of HTFSS in Clinical Diagnosis and Prediction of Immunotherapy Response

The aim of this study was to investigate the distribution of patients’ HTFSS according to different clinical indicators. This was significantly greater in patients older than 60 years (Figure 3A); consistent with standard logic, older patients were associated with higher cancer risk. Scores were also significantly different between patients with different stages of N (Figure 3C). Post hoc pairwise comparisons revealed that the scores of patients with N3 stage were significantly higher than those of patients with N1 and N2 stage, who reported that the scores were able to predict patients’ regional lymph node involvement; that is, the higher the score, the more severe the regional lymph node involvement of the patients. In the patient cohort, unfortunately, there were no significant differences in the M-stage (Figure 3B), T-stage (Figure 3D), and tumor state scores (Figure 3E). Univariate Cox and Multivariate Cox were then used to evaluate the hazard ratio (HR) of clinical features and HTFSS in the cohort of patients. Univariate Cox results (Figure 3F) showed that stage N, stage T, tumor states, and HTFSS were all significant risk characteristics, in which the HTFSS’s HR was 7.3, while the Multivariate Cox (Figure 3G) results showed that the HTFSS’s HR was 7.52. In conclusion, the results of this study indicate that the HTFSS can be used as an independent clinical feature for the purpose of clinical judgment. Subsequent calibration curves for the Nomogram showed a high overlap between true survival and survival predicted by Nomogram at years 1, 3, 5, and 7, suggesting that HTFSS has excellent clinical predictive value (Figure 3H). A Nomogram can calculate a linear predictor value based on the patient’s HTFSS and then obtain the patient’s survival rates at 1 year, 3 years, 5 years, and 7 years (Figure 3I). In order to improve the clinical practicality of HTFSS, we built an interactive Shiny Web page (http://43.138.139.97:3838/sample-apps/HTFSS/ accessed on 14 March 2023) based on the underlying formula of the Nomogram (Figure 3J). Patient survival rates can be calculated by capturing the expression levels of four housekeeping genes and three Hub TF genes, which can help physicians improve the efficiency of clinical diagnosis.

We then also assessed the predictive value of HTFSS for immunotherapies. First, the Estimation of STromal and Immune cells in MAlignant Tumour tissues using Expression data (ESTIMATE) results revealed that low HTFSS levels had a higher immune score (Figure 4A). HTFSS had a significant negative correlation with the immune score (Figure 4B), suggesting that high HTFSS levels have a poor TIME. We also explored the relationship between HTFSS and infiltrating immune cells. Immune infiltration results showed that the level of most immune cells in the High-Score group was low (Figure 4C). Pearson correlation analysis (Figure 4D) showed that HTFSS had a significant negative correlation with 25 types of immune cells, with Activated CD4 T cells having the highest correlation at −0.41. The results of the tumor immunogenicity score (TIGS) evaluation showed that the low HTFSS levels had a higher TIGS (Figure 4E). HTFSS was significantly negatively correlated with TIGS (Figure 4F), suggesting that the higher the HTFSS, the worse the patients’ immunotherapeutic efficacy. In conclusion, the HTFSS immunity predictive value was validated in the paclitaxel-based combination immunotherapy cohort, which showed higher HTFSS in the group that did not respond to treatment. This suggests a negative correlation between HTFSS and immunotherapy outcomes (Figure 4G). In summary, the higher the HTFSS, the lower the infiltration of immune cells, and the worse the effect of immunotherapy on the patient.

### 2.4. The Specific Relationship between HTFSS and TIME Was Analyzed by scRNA-Seq

In this study, we performed low-resolution dimensionality reduction and cluster analysis on scRNA-seq cohorts of 10 patients with TNBC. We obtained 12 major cell clusters (Figure 5A), consisting of T cells, B cells, Natural Killer (NK) cells, plasma cells, macrophages, MKI67+ progenitor cells, endothelial cells, mesenchymal cells, fibroblasts, neutrophils, and epithelial/cancer cells. Then, the individual clustering analysis of lymphoid immune cells (T cells, B cells, NK cells, and plasma cells), macrophages, and epithelial/cancer cells was carried out by high-resolution dimensionality reduction. The lymphoid immune cell was further subdivided into nine major cell clusters (Figure 5B), consisting of plasma cells, T helper cells, Natural Killer T (NKT) cells, exhausted CD8+ T cells, NK cells, regulatory T (Treg) cells, memory T cells, naive B cells, and Plasmacytoid dendritic cells (PDC). Further subdivided into macrophage clusters (Figure 5C) were M1 macrophage, M2 macrophage, tumor-associated macrophage, MKI67+ progenitor cell, granulosa cell, and neutrophil. The CopyKAT algorithm is able to distinguish between normal and cancerous epithelial cells in the case of epithelial/cancer cells (Figure 5D). Lastly, we obtained clusters of 21 different cell types from 10 TNBC patients (Figure 5F). The heatmap of expression (Figure 5E) depicts the specificity of marker gene expression across 21 different cell types. The list of marker genes is shown in Appendix A.

Then, we investigated the expression of three Hub TF genes within the scRNA-seq cohort. The results showed that both *E2F8* (Figure 6A) and *FOXM1* (Figure 6D) were strongly expressed in MKI67+ progenitor cells, while *MYBL2* (Figure 6G) exhibited a pattern of elevated expression in PDCs. When examining the Pearson correlations of the marker genes and the Hub TF genes in these two cells, it found that *E2F8* was associated with MKI67 at 0.5 (Figure 6B) and *E2F8* was associated with *RRM2* at 0.53 (Figure 6C). *FOXM1* and MKI67 had a correlation coefficient of 0.44 (Figure 6E), and *FOXM1* and *RRM2* had correlation coefficients of 0.36 (Figure 6F). The correlation coefficient between *MYBL2* and IRF8 was 0.62 (Figure 6H), and between *MYBL2* and PLAC8 was 0.67 (Figure 6I). The DEGs of *E2F8* and *FOXM1* are extracted on the basis of the MKI67+ progenitor cell queue (Appendix A). Our results demonstrate that *E2F8* is significantly enriched in the PI3K-Akt signaling pathway, Oxidative phosphorylation, and other signaling pathways (Appendix A). *FOXM1* is significantly enriched in Carbon metabolism, Cholesterol metabolism, and the FOXO signal pathway (Appendix A). *MYBL2* extracted DEGs (Appendix A) from the PDC queue, and enrichment results showed *MYBL2* to be significantly enriched in the cell cycle, metabolism of Pyruvate, and other pathways (Appendix A). In summary, *E2F8* and *FOXM1* are hypothesized to be related to MKI67+ progenitor cells and the function of cell proliferation. At the same time, *MYBL2* is linked to PDC and cell cycle functions, and this may be one of the reasons that led to the poor prognosis of three Hub TF genes.

### 2.5. Virtual Docking of Hub TF Genes with Targeted Drugs

Consistent with the differential expression results for the three Hub TF genes in TNBC (Figure 7A), we conducted a screening for antitumor drugs. We downloaded and screened for small-molecule compounds that were able to reduce the mRNA expression of the three Hub TF genes based on the CTD. Interestingly, Calcitriol showed the benefit of high binding energy among small-molecule targeted compounds that reduced the mRNA expression of all three Hub TF genes. Of these, the binding energy for docking between Calcitriol and *E2F8* was found to be −10.63 kcal/mol (Figure 7B), the docking binding energy between Calcitriol and *FOXM1* was −14.75 kcal/mol (Figure 7C), and the binding energy of the docking site between Calcitriol and *MYBL2* was −17.09 kcal/mol (Figure 7D). Calcitriol can thus be listed as one of the crucial targets for potential targeted drugs. We also selected additional drugs that exhibited high docking binding potential and significantly reduced mRNA expression in order to provide avenues of research for combination targeted therapy. In docking with *FOXM1*, DOX exhibited an excellent binding effect, and the binding energy of the docking was −10.06 kcal/mol (Figure 7E). Of the small-molecule compounds targeting *MYBL2*, the *TSA* is notable for having a docking binding energy of −11.63 kcal/mol (Figure 7F). We conclude by proposing a targeted combination treatment plan for the poor prognosis caused by the three Hub TF genes, which provides more options for the individualized treatment of patients.

### 2.6. Research on the Prognostic Function of Hub TF Target Genes

To understand the cause of the poor prognosis caused by Hub TF genes, it is useful to study the imbalance in the activity of pathways affected by downstream transcription factor target genes(Appendix A). The Gene set variation analysis (GSVA) results of the pathways enriched by target genes showed that the Wnt signal pathway, PI3K-Akt signal pathway, FOXO signal pathway, and mTOR signal pathway were upregulated in the three Hub TF genes’ high expression groups. This suggests that three Hub TF genes are upregulated in TNBC cells, and their downstream target genes will affect the positive regulation of functions related to cancer cell proliferation. Furthermore, following the overexpression of *E2F8*, the nucleotide metabolism process, transcription regulator complex, primary miRNA processing, and other pathways related to translation were significantly upregulated (Figure 8A). Following the upregulation of *E2F8*, it is hypothesized that its target genes promote the proliferation and differentiation of cancer cells by affecting the RNA translation process. Interestingly, the activity of the group with high *FOXM1* expression was significantly upregulated during DNA replication, such as DNA replication, DNA geometrical change, and signaling through the mitotic intra-S DNA damage checkpoint (Figure 8B). We hypothesize that cancer cells in TNBC may enhance DNA replication activity by positively regulating *FOXM1* expression and its downstream target genes, thereby promoting cancer cell occurrence and development. The cell cycle-related pathway activity was significantly upregulated in the group with high *MYBL2* expression, such as regulation of the cell cycle G1/S phase transition, positive regulation of the cell cycle, and mitotic cell cycle phase transition (Figure 8C). We hypothesize that *MYBL2′*s downstream target gene could affect cancer cell proliferation and differentiation by promoting cell cycle progression.

### 2.7. DOX Combined with TSA Significantly Inhibited the Proliferation of TNBC Cells

After treating *MDA-MB-231* cells with DOX, TSA, and Calcitriol at different concentrations, the CCK-8 assay showed that both DOX and TSA had varying degrees of cytotoxic effects on *MDA-MB-231* cells (Figure 9A,B). The IC_50_ value of DOX for *MDA-MB-231* cells was about 500 ng/mL, the IC_50_ value of TSA for *MDA-MB-231* cells was about 0.6 μM, and the antitumor effect of Calcitriol was not significant (Figure 9C). TSA of 0.6 μM was combined with DOX at a range of concentrations (Figure 9D), and TSA resulted in a significant reduction in DOX dose. In particular, when the DOX dose was 125 ng/mL, the combination of DOX and TSA achieved nearly the same effect as DOX at a dose of 500 ng/mL. Consistent with the results of the previous analysis, the expression levels of three Hub TF genes were found to be highly upregulated in TNBC. To further confirm that all three Hub TF genes were significantly upregulated in the *MDA-MB-231* cells, RT-qPCR experiments were performed at the cellular level (Figure 9E–G). After *MDA-MB-231* cells were treated with DOX (250 ng/mL) and TSA (0.6 μM), the expression of both *FOXM1* and *MYBL2* was downregulated, and the expression of *E2F8* was upregulated in the presence of DOX treatment (Figure 9H,J).

The results of the preliminary analysis showed that the elevated expression of *FOXM1*, *E2F8*, and *MYBL2* in TNBC patients was able to promote the activities of the PI3K-AKT and FOXO signaling pathways, and DOX and TSA were able to effectively reduce the expression levels of all three transcription factors. Therefore, we further explored the relationship of DOX and TSA to the aforementioned pathways. Cells were treated with DOX (250 ng/mL) and TSA (0.6 μM) for 48 h in the presence of *MDA-MB-231*. Based on Western blotting detection results, DOX and TSA were shown to significantly decrease *mTOR* and *PI3K* protein expression levels and the levels of phosphorylation of *P70*, *S6*, and *4EBP1* within the PI3K-AKT signaling pathway (Figure 9K). In the FOXO signaling pathway, the phosphorylation level of the *FOXO1* protein was decreased, while that of the *FOXO3* protein was increased (Figure 9L). More significant protein changes were observed, particularly after DOX was combined with TSA. Based on these findings, it was hypothesized that DOX and TSA could inhibit tumor cell progression by regulating PI3K-AKT and FOXO signaling pathways.

### 2.8. The Combination of DOX and TSA Significantly Inhibited the Progression of TNBC in Mice, and Calcitriol Could Improve the Cardiac Function of Mice

In addition, we used an orthotopic *mouse* tumor model to test the effects of all three drugs on tumor growth in *mice*. The results showed that, compared to the control group of *mice* receiving intraperitoneal TSA or DOX injections alone, TSA and DOX, or TSA, DOX, and Calcitriol were all capable of inhibiting TNBC tumor progression to some degree. Of these, the DOX and TSA + DOX combination groups were found to have more significant effects (Figure 10A–C). The results of the changes in *mouse* weight showed that *mice* in the DOX group had a significant decrease in weight. At the same time, the combination of TSA and DOX, as well as the combination of TSA, DOX, and Calcitriol, was able to alleviate the weight loss caused by DOX. The combination of all three drugs had the largest effect size (Figure 10D). TSA and Calcitriol may reduce DOX’s toxic side effects in *mice* to some extent. The expression of *ki67*, a tumor proliferation-associated protein, was examined by immunohistochemistry (Figure 10E). *Ki67* protein expression in tumor tissues was significantly decreased in the TSA group, the DOX group, the TSA + DOX combination group, and the combination TSA, DOX, and Calcitriol treatment group compared to the control group. Of these, the DOX group had the largest effect size. Consistent with the analysis of HE staining in *mouse* cardiac tissues (Figure 10F), we found that there were extensive inflammatory aggregates and damage to myocardial cells in cardiac tissues from *mice* in the DOX group. On the other hand, after treatment with the combination of DOX, TSA, and Calcitriol, the cardiac toxicity caused by DOX in the *mice* was significantly improved. Of these, the combination of all three drugs had the greatest protective effect on the heart, indicating that Calcitriol was able to improve cardiac function in *mice*.

## 3. Discussion

TNBC is among the most prevalent female cancers worldwide, with the hallmarks of high deterioration, high invasiveness, high heterogeneity, and a poor prognosis. As the largest branch of the artificial intelligence field, machine learning has invaluable opportunities in the diagnosis and prediction of TNBC [15]. As critical factors in complex regulatory networks, TFs regulate the expression of a variety of genes. Breast cancer TF research can guide targeted breast cancer diagnosis and therapy [15,16]. Currently, the transcription factor SRY box 10 (*SOX-10*) and the type 1 trichorhinophalangeal syndrome (*TRPS-1*), as markers of breast cancer diagnosis by IHC, have been included in TNBC [17] diagnostic assays. *BACH1* regulates cancer cell invasion genes, aiding in the migration and invasion of breast cancer cells, whereas acetyl tricyclic bis (cyanoketone) can produce novel activity and destroy *BACH1* expression [18]. The pressing need for novel biomarkers is also increasing as more patients are diagnosed with TNBC. TFs specific to TNBC may help improve the rate of diagnosis and provide a targeted therapeutic target.

The purpose of this study was to use machine learning to construct HTFSS highly correlated with TNBC, which was made up of the ratio of four housekeeping genes (*ACTB*, *GAPDH*, *TFRC*, and *TUBB*) and three TFs (*E2F8*, *FOXM1,* and *MYBL2*), which had an excellent ability to predict prognosis. In the training and validation cohorts, the AUC values at 1 year were 0.877 and 0.987, respectively. Similarly, the results of the Cox regression analysis showed that the HR for the HTFSS was much greater than one, which may serve as an independent clinical risk factor. Applying HTFSS to different clinical features of TCGA, HTFSS was found to be significantly correlated with the N stage of the tumor. This means that the higher the HTFSS, the worse the patient’s lymph node involvement. Studies of immune infiltration and immunotherapy cohorts have also shown that HTFSS positively correlates with poor immunotherapy outcomes. This means that the higher the HTFSS, the worse the patient’s immunotherapeutic effect. We conclude that the HTFSS developed by us based on TFs has a great deal of clinical prognostic value, which may not only predict patient prognosis and lymph node involvement but also the effectiveness of immunotherapies. More importantly, we have developed a Nomogram-based online interactive Web page that can be applied to clinical diagnostics by detecting the expression levels of only four housekeeping genes and three genes encoding TFs, greatly enhancing the translation of clinical research. Subsequent studies demonstrated that HTFSS affected the occurrence and development of TNBC and TIME through the regulation of pathways related to MKI67+ progenitor cell and PDCs proliferation. To suppress the prognostic issues caused by the imbalance in HTFSS, we developed a combination strategy and screened three targeted drugs, DOX, TSA, and Calcitriol, by molecular docking and studied their antitumor effects. We found that DOX and TSA effectively inhibited *E2F8*, *FOXM1,* and *MYBL2* overexpression and the abnormal expression of PI3K-AKT and FOXO signaling pathways. DOX + TSA combination therapy can effectively reduce DOX dosing. The combination of DOX + TSA + Calcitriol can effectively reduce cardiotoxic side effects caused by DOX while ensuring the antitumor effect, which undoubtedly led to the optimization of the clinical treatment plan for TNBC patients. In summary, studies on the prognostic mechanism and the combination of targeted drugs based on HTFSS provide a new perspective on the molecular mechanism and clinical treatment of HTFSS. However, there are also some ethical considerations when using HTFSS for clinical applications. Undeniably, the machine learning model has some limitations as well. Thus, HTFSS guidance for clinical diagnosis can only be used as a reference and cannot be relied upon for the final conclusion. Other clinical indicators must be considered to determine the specific drug response and prognosis of patients.

The prognostic molecular mechanisms of the three Hub TF genes were explored from the scRNA-seq and Bulk RNA-seq cohorts, respectively. *E2F8* is a member of the E2F transcription factor family that regulates gene expression required for cell cycle progression [19], and *E2F8* is involved in the development and progression of various cancers [20,21]. The results of the scRNA-seq analysis show that *E2F8* is related to the MKI67+ progenitor cell’s function. A high expression of *MKI67* characterizes MKI67+ progenitor cells. This gene encodes a nuclear protein necessary for cell proliferation that has been involved in various types of carcinogenesis. Further pathway enrichment analysis showed that *E2F8* controlled the cell cycle of MKI67+ progenitor cells by regulating the PI3K-Akt signaling pathway and Oxidative phosphorylation. The results of downstream target genes in Bulk RNA-seq showed that the activity of translation-related pathways was significantly upregulated after overexpression of *E2F8*, suggesting that *E2F8* promotes cancer cell proliferation and differentiation by affecting the translation process of RNA. It is speculated that *E2F8* is upregulated in TNBC, which affects the proliferation and development of MKI67+ progenitor cells and tumor cells, as well as the poor prognosis of patients. *FOXM1* is a transcriptional activator in cell proliferation that regulates DNA replication and DNA fragmentation repair [22]. *FOXM1* is also related to MKI67+ progenitor cell functionality. *FOXM1* promotes the proliferation of MKI67+ progenitor cells by participating in Carbon metabolism, Cholesterol metabolism, and the FOXO signal pathway. The study of downstream target genes found that *FOXM1* upregulated can also promote the activity of DNA replication. It is speculated that the high expression of *FOXM1* will affect the proliferation of MKI67+ progenitor cells and tumor cells, resulting in a worse prognosis. As one of the members of the *MYB* family, the proteins encoded by *MYBL2* are also nuclear proteins involved in the cell cycle process, which plays an essential role in regulating cell survival, proliferation, and differentiation [23,24]. The results of the scRNA-seq analysis show that *MYBL2* is related to PDC functionality. PDC is a unique subset of dendritic cells with typical myeloid and lymphoid cell characteristics. It has the particular function of secreting type I IFN, which enables them to quickly secrete large amounts of IFNα and β to deal with viral infection [25,26]. Further pathway enrichment analysis showed that *MYBL2* was related to metabolic pathways such as the cell cycle and Pyruvate metabolism. Bulk RNA-seq analysis showed that after the high expression of *MYBL2*, the activity of cell cycle-related pathways was also significantly upregulated. It is speculated that the proliferation of cancer cells caused by *MYBL2* will lead to a relative decrease in PDC and other immune cells. The resulting immunodeficiency further destroys the immune regulation in the tumor microenvironment, making the tumor obtain immune escape and leading to a worse prognosis for the patients. It is worth noting that after *E2F8*, *FOXM1*, and *MYBL2* were upregulated, the activities of the PI3K-Akt, Wnt, FOXO, and mTOR signaling pathways were all upregulated. As a pivotal pathway to promote cancer progression, elucidating how three Hub TF genes and downstream target genes affect the above four pathways is the top priority of the following study.

Based on the upregulation of three Hub TF genes in TNBC, combined with the CTD and Autodock molecular docking, three clinically commonly used targeted drugs (DOX, TSA, and Calcitriol) were screened, and a new scheme for combined therapy was proposed. Calcitriol is a synthetic physiologically active Vitamin D analog that can induce cell cycle arrest in the G0/G1 phase, promote cell differentiation, and induce apoptosis, thereby inhibiting the proliferation of tumor cells. It is reported that Calcitriol can be used to treat advanced prostate cancer, hypercalcemia, and hyperphosphatemia [27,28]. DOX is an anthracycline antibiotic that prevents DNA replication and inhibits protein synthesis from producing antitumor activity. It is one of the current hot spots for antitumor drugs [29,30]. Additionally, DOX is frequently used as an immunologic adjuvant in antitumor immunotherapy [31]. TSA is a natural derivative isolated from Streptomyces. That binds explicitly to histone deacetylation, leading to over-acetylation of core histones, promoting selective gene transcription, and inhibiting tumor growth [32]. In addition, TSA increased the sensitivity of liver cancer cells to paclitaxel [33]. This study proved that TSA had a noticeable killing effect on TNBC cells in vitro, and the combination of TSA and DOX significantly reduced the dose of DOX. The results of Bulk RNA-seq downstream target gene studies showed that the high *FOXM1*, *E2F8*, and *MYBL2* expression in TNBC patients could upregulate the activity of the PI3K-AKT and FOXO signal pathways. RT-qPCR and Western blotting experiments showed that after administration of TSA, the expression of *FOXM1* and *MYBL2* in TNBC cells was downregulated, and the expression of *E2F8* almost returned to the normal level. It inhibited the proliferation of tumor cells through the PI3K-AKT and FOXO signal pathways. The expression of proliferation-related protein *ki67* in the tumor was detected by orthotopic tumor transplantation in nude *mice* and immunohistochemistry, which proved that TSA could inhibit the progression of TNBC tumors. In vivo and in vitro experiments showed that Calcitriol had no pronounced antitumor effect, but it could reduce DOX’s toxic and side effects on *mice*. The changes in body weight and HE staining of heart tissue showed that the combined group of TSA + DOX + Calcitriol had the most significant protective effect on the heart. This indicates that Calcitriol could protect the cardiomyocytes from the toxic damage of DOX and improve the cardiac function of *mice*. To sum up, TSA has an anti-TNBC effect, and the antitumor effect of TSA combined with DOX is twice as good as that of DOX alone. Calcitriol can reduce the toxic effects and side effects of DOX on *mice* and improve their cardiac function. In addition, our research shows that screening multiple targeted drugs for oncogenes is feasible to guide the combination of drugs. In the future, predicting the efficacy of combination drugs through deep learning can further help predict the combined use of multiple targeted drugs. Furthermore, there has been widespread concern about new biological materials in the field of targeted drugs, including hyaluronic acid hydrogel materials and nanomaterials [34,35], which are characterized by the efficient delivery of drugs to the target area with a minimum of biotoxicity in order to achieve maximum efficacy [36]. In terms of implementing the targeted combination drug therapy strategy reported in this study in a clinical setting, this could be a viable option for considering the use of new biomaterials to construct drug combination models for drug delivery.

Lastly, this study has some limitations. The sample sizes of both Bulk RNA-seq and scRNA-seq cohort studies were small, leading us to ignore some information. Thus, in future joint analyses of multiple omics, we can consider using the strategy of multi-sample integration to broaden the search scope. Second, this study revealed that dysregulation of HTFSS can affect changes in the activity of four key pathways. The precise molecular mechanism remains to be explored, and further experimentation is required to verify these findings, which will be the focus of future work. Furthermore, the combination drug strategy remains in the preliminary exploratory stage, and further optimization is required in future research work, such as dose control of combination drugs and assembly of targeted drug delivery.

## 4. Materials and Methods

### 4.1. Data Sources

We downloaded the TCGA-BRCA [37] from UCSC XENA and used it for differential analysis, WGCNA, and the construction of HTFSS. GSE58812 and GSE173839 were used as validation cohorts, and scRNA-seq cohorts (GSE176078) for further analysis of Hub TF genes were downloaded from the Gene Expression Omnibus database [38].

### 4.2. Selection of TFs in Hub Gene Sets Highly Related to TNBC

The patients with TCGA-BRCA were divided into tumor and normal groups for differential analysis by the DEseq2 package [39]. A threshold was set to filter DEGs: Cutoff for log2FoldChange = 2 and *p*-value < 0.05. Using DEGs as the input of WGCNA [40], the gene sets were classified into modules through cascade system clustering and correlation networks. Then, TCGA-BRCA phenotypes were analyzed through the TCGAbiolinks package [41] in R, which drew a heatmap to show the correlation between phenotypic information and gene modules and selected significant positive correlation modules for further analysis. Next, we calculated phenotypic information and the value of module membership (MM) and gene significance (GS) genes within the selected module and used them to screen Hub genes. The selection criteria were MM > 0.8 and GS > 0.2. Finally, we downloaded the TF set in the database of AnimalTFDB [42,43], took the intersection of the TF and Hub genes, and used the Hub TF genes in the intersection set for further analysis.

### 4.3. Construction and Validation of the HTFSS

LASSO regression [44] is a machine learning model training method. It mainly constructs a penalty function and uses the L1-regularization method to continuously compress the regression coefficients to screen the prediction variables, thus obtaining an acceptable and easy-to-interpret model. For the applicability of the HTFSS, before LASSO regression, we standardized the data by dividing the expression values of predictive variables in the training cohort by the expression values of four housekeeping genes (*ACTB*, *GAPDH*, *TFRC*, and *TUBB*). After the HTFSS was constructed, the predictive performance of the scoring system was evaluated by the AUC value of the area under the time-dependent ROC curve.

### 4.4. Explore the Clinical Application Value of HTFSS and the Predictive Value of Immunotherapy

Univariate Cox and Multivariate Cox regression were used to study the relationship between HTFSS and clinical features. A Nomogram [45] was constructed to help understand the relationship between HTFSS and patient survival. To further expand the utility of the model, we built the Shiny App based on the underlying formula of the Nomogram. Subsequently, ESITIMATE, single-sample gene set enrichment analysis (ssGSEA), and TIGS algorithms were used to study the relationship between HTFSS and immune cell infiltration and immunotherapy outcomes. In the ESTIMATE algorithm, exploring the immune and stromal scores of HTFSS can illustrate the relationship between HTFSS and TIME. In addition, the relationship between HTFSS and the abundance of 28 immune cells was calculated by the ssGSEA algorithm [46] combined with the correlation analysis. TIGS is a critical factor in determining whether a tumor is clinically responsive to immunotherapy, which combines the expression characteristics of tumor mutational burden (TMB) and antigen processing presentation mechanism (APM). The GSVA algorithm assessed APM and normalized: APM_normalized_ = APM − APM_cancer_min_ × APM_cancer_max_ − APM_cancer_min_; TIGS = APM_normalized_ × ln(TMB + 1).

### 4.5. scRNA-Seq Cohort Research

The scRNA-seq cohort was studied by the Seurat package (v4.0) in R. The mitochondrial gene content (percent. mt) was set to <15% to filter the single-cell matrix. The matrix was normalized; PCA dimensionality reduction was passed through the SNN algorithm (resolution = 0.4) to cluster cell clusters and display single-cell clusters with dimensionality reduction through the UMAP algorithm. Then, the FindAllMarkers function (min.pct = 0.35, logfc.threshold = 0.5) was used to find the marker genes of all cell clusters. The cell clusters were annotated in a semi-supervised manner, that is, automated annotation through SingleR first, followed by CellMaker and PanglaoDB [47,48] databases, which were judged to confirm the final cell population. In addition, the CokyKAT algorithm was used to distinguish cancer cells from Mixed epithelial and cancer cell clusters. The grouping criteria for Hub TF genes in different cell cohorts were as follows: the cells with the first 30% expression level were defined as the high expression group, and the cells with the last 70% expression level were defined as the low expression group. The Limma package [49] performed differential analysis on the two groups. The clusterProfiler package performed KEGG or GO pathway enrichment analysis on the obtained DEGs.

### 4.6. Targeted Drug Screening of Hub TF Genes

We downloaded the collection of targeted small molecule compounds from the CTD [50], the 3D structures of these small molecule compounds from the PubChem Database [51], and the 3D structures of gene expression products from the Uniport Database [52]. Autodock (v4.2, Linux) is used to simulate molecular docking, and the lowest docking binding energy can determine the targeting drug with the best binding degree to a protein. At last, the docking results are visualized by the open-source version of Pymol (v2.6).

### 4.7. Research on the Prognostic Function of Hub TF Target Genes

In Bulk RNA-seq, the DEGs of the Hub TF gene were calculated using the scRNA-seq grouping method. Significant co-expressed genes were screened using Pearson correlation. Subsequently, the target genes of the Hub TF gene were downloaded from the databases of GTRD [53], Cistrome [54], and hTFtarget [55], respectively. In order to reduce the number of target genes, the intersection between every two target gene sets is first taken, and then these intersections are combined with the intersections of the three sets. Finally, to accurately locate the target gene of Hub TF, the combined target gene sets were intersected with DEGs and co-expressed genes. The final result obtained was defined as the target gene of Hub TF. The prognostic dysfunction caused by Hub TF target genes was investigated by comparing the KEGG/GO pathway activity enriched by target genes between the high- and low-expression groups.

### 4.8. In Vitro Experimental Verification

#### 4.8.1. Cell Culture

Human triple-negative breast cancer cells MDA-MB-231 were cultured in DMEM medium (Gibco, Thermo Scientific, Waltham, MA, USA, C11995500BT) supplemented with 10% fetal bovine serum (Procell, Wuhan, China, 164210-500) and 1% penicillin/streptomycin and incubated at 37 °C and 5% CO_2_. The medium was changed every 24 h, and experiments were performed with logarithmically growing cells. *TSA* (HY-15144) and Calcitriol (HY-10002) were purchased from MCE.

#### 4.8.2. CCK-8

*MDA-MB-231* cells were cultured to the logarithmic growth stage and treated with DOX, TSA, and Calcitriol at different concentrations in 48-well plates. Four groups were set up: the DOX group, the TSA group, the Calcitriol group, and the TSA and DOX group. After 48 h of administration, the cell medium was removed, and the cells were purified twice with 1x PBS. Complete medium and 10% detection reagent were added to the cells, and the cells were incubated at 37 °C for 30 min in the dark. The OD values of each well were detected with an enzyme marker at a 450 nm wavelength.

#### 4.8.3. Real-Time q-PCR (RT q-PCR)

RNA was extracted from cells. We collected the cell precipitates and added 1 mL of Trizol to lyse the cells for 5 min on ice. We added 200 μL of trichloromethane, mixed it by shaking, and placed it on ice. When the solution forms stratification, transfer it to a low-temperature centrifuge and centrifuge at 12,000 rpm for 10 min at 4 °C. The solution will form three RNA, DNA, and organic phase layers. Then, we transferred the upper RNA layer to a new 1.5 mL EP tube and added 0.8 times the volume of isopropyl alcohol. After inverted mixing, we stored the solution in the refrigerator at −20 °C for 30 min and then centrifuged again at a low temperature and 12,000 rpm for 10 min. The supernatant and purification of the RNA precipitation with 75% and 95% ethanol were discarded. After the remaining ethanol was absorbed and volatilized, we added 30–50 μL of DEPC water to dissolve the RNA precipitation. The quality and concentration of the extracted RNA were detected.

cDNA synthesis by reverse transcription. The RNA was converted into cDNA using the Hifair III 1st Strand cDNA Synthesis SuperMix for qPCR (Yeasen Biotechnology (Shanghai) Co., Ltd., Shanghai, China) reverse transcription kit. The system for reverse transcription was configured according to the instructions provided by the kit. After the run, the resulting cDNA was used for subsequent RT-qPCR experiments.

RT-qPCR. Gene expression was analyzed using the FastStart Universal SYBR Green Master (Roche) test kit. The system was configured according to the instructions provided by the kit, with three replicate wells per sample. After sampling, the 96-well plates were put into the fluorescence quantitative PCR instrument for computer detection. *GAPDH* was used as an internal reference gene. Data were calculated using the 2^−∆∆Ct^ method. Primer sequences for the Hub TF genes are shown in Appendix A.

#### 4.8.4. Western Blotting Analysis

The *MDA-MB-231* cells were treated with DOX (250 ng/mL) and TSA (0.6 μM) for 48 h. After treatment, the cells were collected and lysed in RIPA buffer to extract the protein. The protein concentration was determined using the BCA protein quantification method. Equal amounts of protein were subjected to electrophoresis with a 10% SDS-PAGE gel (140 V, 200 mA, 1 h) and transferred to a PVDF membrane (200 V, 300 mA, 3 h). The membrane was blocked with 5% skim milk or 2% BSA for 2 h and incubated overnight with the primary antibodies. These antibodies include *GAPDH* (ABclonal, AC001, 1:1000), *β-Actin* (Abclonal, AC026, 1:1000), *Vinculin* (Bioss, bs-6640R, 1:1000), *PI3K* (CST, 4257S, 1:1000), *AKT* (CST, 4691S, 1:1000), *mTOR* (CST, 2972S, 1:1000), *P70* (CST, 2708S, 1:1000), *S6* (CST, 2217S, 1:1000), *4EBP1* (CST, 9452S, 1:1000), *p-P70* (CST, 9234S, 1:1000), *p-S6* (CST, 5364S, 1:1000), *p-4EBP1* (CST, 9451S, 1:1000), *FoxO1* (Millipore, 05-1075, 1:1000), *FoxO3* (CST, 12829P, 1:1000), *p-FoxO1* (CST, 9464P, 1:1000), *p-FoxO3* (CST, 9465P, 1:1000). Then, the membranes were washed with Tris-buffered saline and Tween-20 for 10 min and incubated with secondary antibodies for 2 h at room temperature. The developer solution was prepared according to the horseradish peroxidase (HRP) instructions. The chemiluminescence signals were detected with enhanced chemiluminescence (ECL).

### 4.9. In Vivo Tumor Model

#### 4.9.1. Construction of Orthotopic Transplanted Tumor in Nude Mice

Clean female BALB/C nude *mice* were purchased from Yunnan University, and adriamycin was purchased from Pu ‘er People’s Hospital. The BALB/C nude *mice* were kept in SPF houses, and their drinking water was changed every 2 days. Their food was added every 5 days, and their bedding was changed weekly. Twenty BALB/C nude *mice* were randomly divided into five groups: Control group, TSA group, DOX group, TSA + DOX group, and TSA + DOX + Calcitriol group. After one week of adaptive feeding, *MDA-MB-231* tumor cells were mixed with matrix glue, and nude *mice* (1 × 10^6^/cell) were injected in situ under the skin to construct the nude *mouse* orthotopic transplantation tumor model. After tumor formation, the nude *mice* were treated with intraperitoneal injections of DOX (2 mg/kg), TSA (1 mg/kg), and Calcitriol (0.5 μg/kg) once every three days, while the control group was treated with the same dose of normal saline. The changes in body weight and tumor size were recorded every three days during feeding. Twenty-one days after administration, the *mice* were killed by neck dislocation, and tumor, heart, and other tissues were collected for follow-up experiments.

#### 4.9.2. Immunohistochemical Staining

Fresh heart tissue was soaked in 4% paraformaldehyde for 24 h. After dehydration, transparency, and paraffin embedding, the paraffin block was sliced into sections of 4 μm thickness. The sections were dewaxed using xylene and an ethanol gradient, followed by inactivation. Next, a 3% H_2_O_2_ solution was added and incubated for 20 min at room temperature, away from light. Antigen repair was performed by boiling the sections in a 2% sodium citrate solution. To seal the sections, 5% goat serum was added. The primary antibody *ki67* (Abcam, AB16667, 1:500) was added and incubated overnight at 4 °C. Next, the sections were incubated at 37 °C for 1 h with the HRP RABBIT/*MOUSE* (Dako, Agilent Technologies, Santa Clara, CA, USA) secondary antibody. DAB was used for the color reaction, followed by post-staining with hematoxylin for 1 min and conversion to blue using ammonia. After dehydration using an alcohol gradient, the sections were sealed with neutral gum, and the tumor tissue was viewed under a microscope.

#### 4.9.3. Hematoxylin and Eosin (H&E) Staining

Fresh heart tissue was soaked in 4% paraformaldehyde for 24 h. After dehydration, transparency, and paraffin embedding, the paraffin block was sliced into sections of 4 μm thickness. The sections were then dewaxed in xylene Ⅰ and xylene Ⅱ for 10 min each and rehydrated using a gradient of ethanol. Hematoxylin was used to stain the sections for 1 min, followed by hydrochloric acid differentiation and ammonia rehydration. Eosin was then used to stain the sections for 30 s, and after dehydration, the slides were sealed with neutral gum. The pathological changes in myocardial tissue were observed under a microscope.

### 4.10. Statistical Analysis

Statistical analysis was mainly performed by R language (v4.1.1) and GraphPad Prism (v8.0.2) in this study. The Wilcoxon test was used for statistical testing between different groups, and the Logrank test was used to test the significant difference in survival probability between samples. All of the experiments were repeated at least three times, with the data presented as means ± standard error of the mean (SEM) using one-way ANOVA to determine significant differences between multiple groups. *p*-value < 0.05 indicated statistical significance.

## 5. Conclusions

In the present study, a machine learning model, HTFSS, which has outstanding prognostic and predictive value for immunotherapy efficacy, was established based on transcription factors. As an independent clinical high-risk factor, HTFSS can be used to predict lymph node involvement in patients with TNBC. A Shiny App based on HTFSS is helpful for clinical diagnosis. The pathway activity changes caused by the imbalance of three Hub TF genes (*E2F8*, *FOXM1*, and *MYBL2*) in HTFSS were discussed from different angles. The three Hub TF genes are highly upregulated in TNBC and affect the proliferation of tumor cells through the PI3K-Akt signaling pathway, Wnt signaling pathway, FOXO signaling pathway, and mTOR signaling pathway, resulting in a poor prognosis. Targeted drug screening and virtual docking based on Hub TF genes provide a new idea for combined drug use. Cellular and *mouse* experiments have proved that TSA has anti-TNBC activity, and the combination of TSA and DOX can effectively reduce the dose of DOX. Although Calcitriol does not have anti-TNBC activity, the combination of TSA and DOX can significantly reduce the cardiac side effects caused by DOX. In conclusion, our study provides a new point of view on the prognostic diagnosis and targeted drug therapy of TNBC.

## Figures and Tables

**Figure 1 ijms-24-13497-f001:**
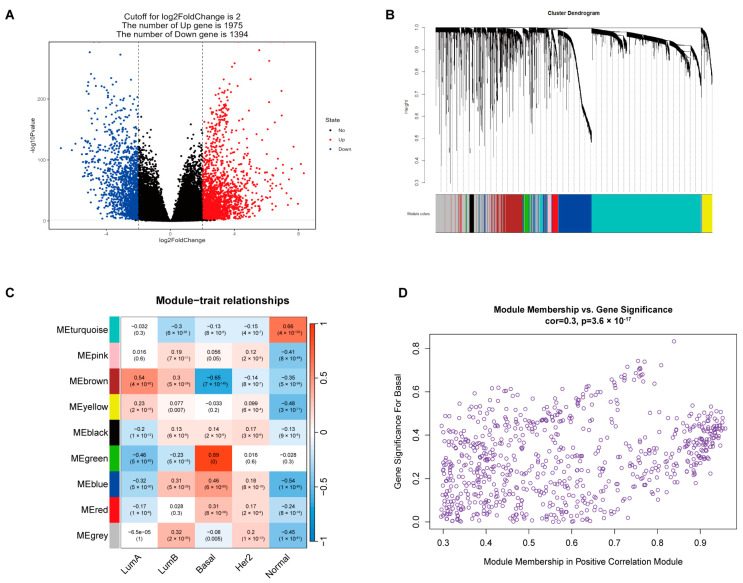
Screening of Hub genes for the triple-negative breast cancer (TNBC) phenotype. (**A**) Up- or downregulated differentially expressed genes (DEGs) were obtained from The Cancer Genome Atlas cohort of breast cancer patients (TCGA-BRCA) tumor and normal subgroups. (**B**) Weighted correlation network analysis (WGCNA) divides DEGs into gene modules of different colors. (**C**) Pearson correlation analysis was performed between different color modules and BRCA clinical phenotypes, and significance (*p*-value) is indicated in parentheses. (**D**) Component and correlation analysis of genes within black, green, blue, and red modules with BRCA-Basal phenotypes.

**Figure 2 ijms-24-13497-f002:**
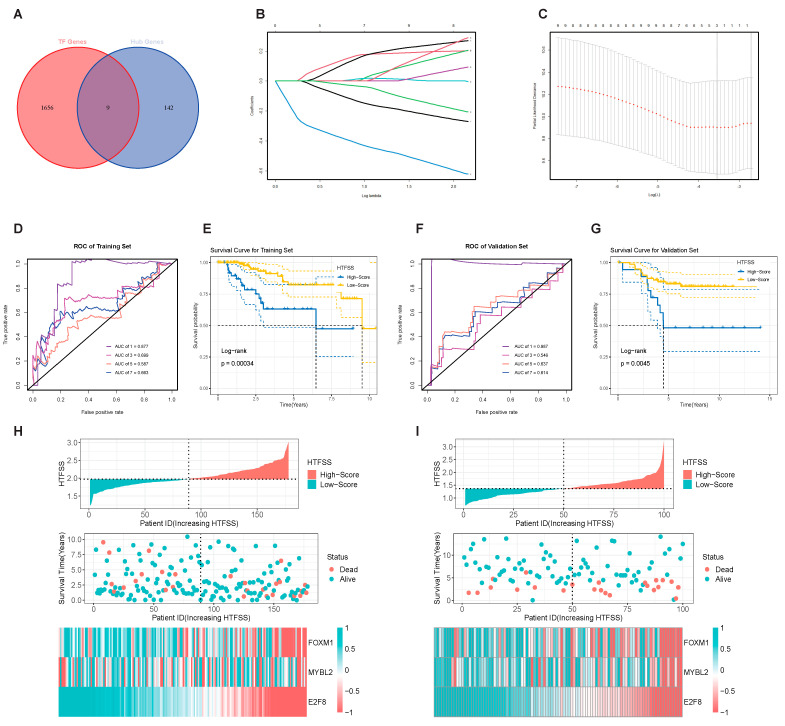
Construction and verification of the machine learning model Hub TF Feature Scoring System (HTFSS) based on Hub TF. (**A**) Venn diagram: the intersection of the Hub gene set and the transcription factors (TFs) gene set. (**C**) Scaling of The Least Absolute Shrinkage and Selection Operator (LASSO) regression parameters and (**B**) screening of genetic variables. The 1-, 3-, 5-, and 7-year Area Under Curve (AUC) values of the HTFSS in the training cohorts (**D**) and validation cohorts (**F**), respectively. The Kaplan-Meier survival curves of patients in the high and low HTFSS groups in the training cohort (**E**) and validation cohort (**G**). Associations between HTFSS, survival status, and prognostic gene expression levels of patients in training cohorts (**H**) and validation cohorts (**I**).

**Figure 3 ijms-24-13497-f003:**
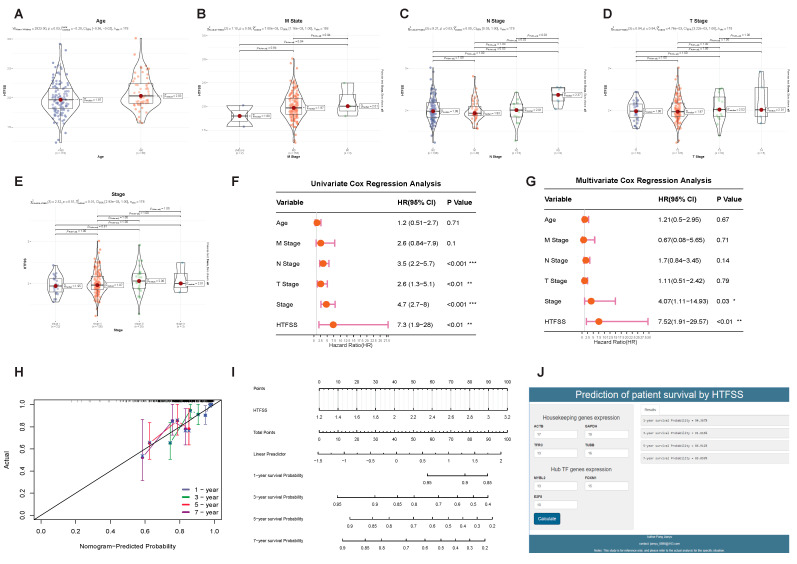
Clinical value evaluation of HTFSS. The patients were grouped according to age (**A**), tumor M stage (**B**), tumor N stage (**C**), tumor T stage (**D**), and tumor status (**E**). The differences in HTFSS among the same clinical characteristic groups were compared. Univariate Cox (**F**) and Multivariate Cox (**G**) were used to analyze the risk ratios of the above clinical characteristics and HTFSS at the same time, and the forest plot was drawn. *p* < 0.05 was a significant difference, * *p* < 0.05; ** *p* < 0.01; *** *p* < 0.001. Under this premise, the hazard ratio (HR) > 1 indicated that the factor had a high-risk factor, and HR < 1 indicated that the factor was a protective factor. The Nomogram (**I**) and its calibration curve (**H**) were constructed. The survival rate of the Nomogram could be calculated according to the HTFSS, and the calibration curve could be used to determine the prediction accuracy of the model. (**J**) Screenshot of the interactive Shiny App web page.

**Figure 4 ijms-24-13497-f004:**
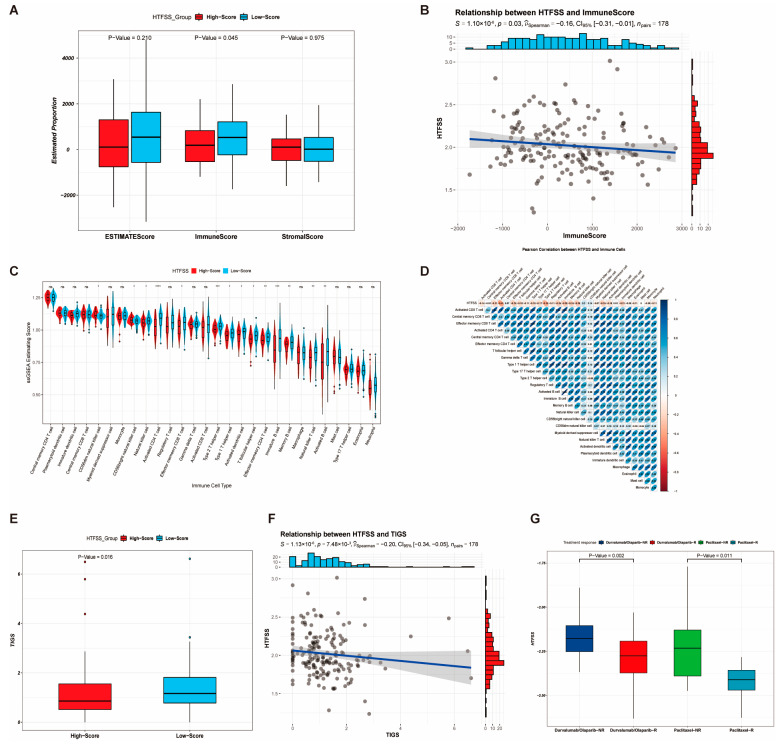
Assessment of the immune predictive value of HTFSS. (**A**) Differences in immune and stromal scores between HTFSS subgroups. (**B**) Correlation analysis of HTFSS and immune scores. (**C**) Abundance differences in 28 different immune cells between HTFSS subgroups, * *p* < 0.05; ** *p* < 0.01; *** *p* < 0.001; **** *p* < 0.0001. (**D**) Correlation analysis of HTFSS and 28 immune cells. (**E**) The tumor immunogenicity score (TIGS) differences between HTFSS subgroups. (**F**) Correlation analysis of HTFSS and TIGS. (**G**) Differences in HTFSS between the response (R) and non-response (NR) groups in the paclitaxel combination immunotherapy cohort.

**Figure 5 ijms-24-13497-f005:**
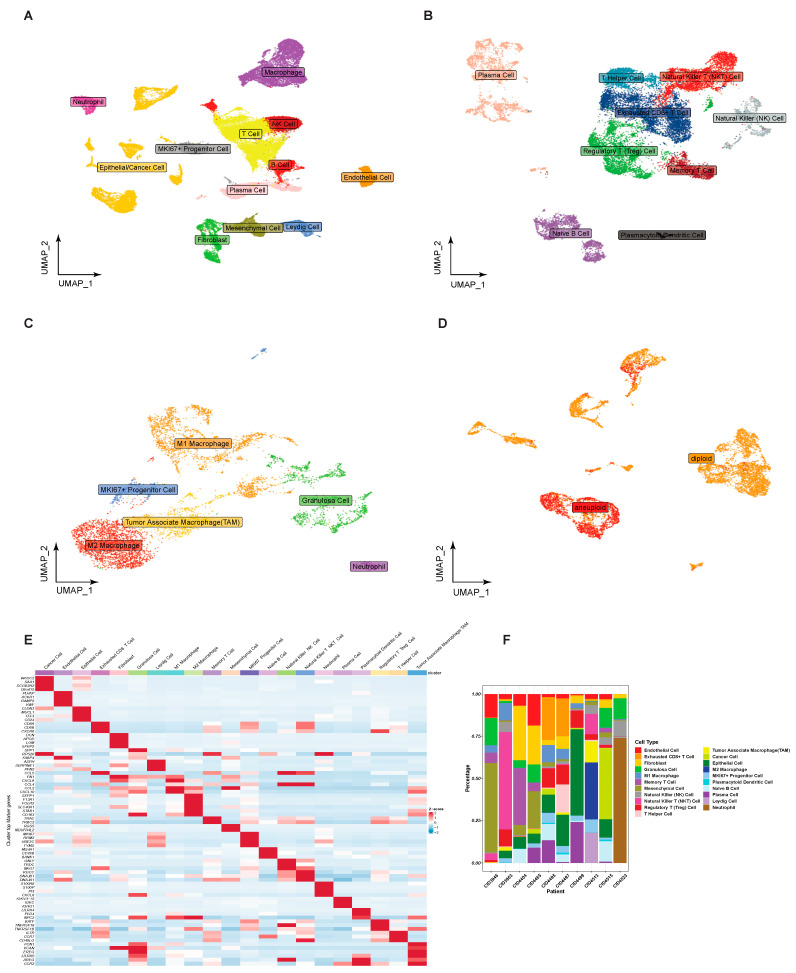
Single-cell RNA-seq (scRNA-seq) cohort analysis of 10 TNBC patients. (**A**) Information on the low-resolution cell clusters identified by UMAP dimensionality reduction and KNN clustering. Further identified the high-resolution cell cluster information by UMAP dimensionality reduction and KNN clustering in lymphoid immune cells (**B**), macrophages (**C**), and epithelial/cancer cells (**D**), respectively. Abundance distribution of 21 different cell types in 10 TNBC patients (**F**) and a heatmap (**E**) of marker gene expression.

**Figure 6 ijms-24-13497-f006:**
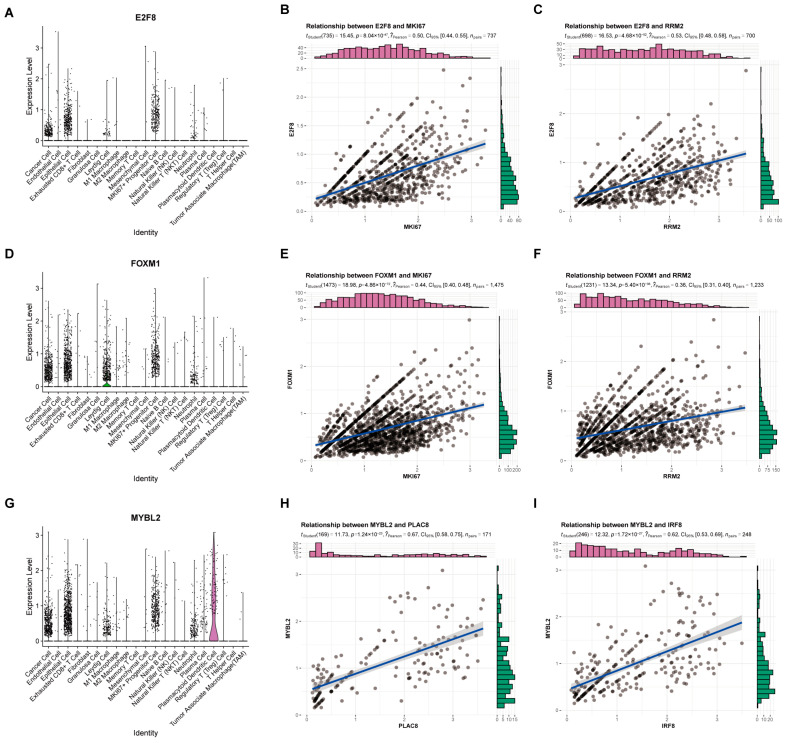
Preliminary exploration of three Hub TF genes in the scRNA-seq cohort. Expression of *E2F8* (**A**), *FOXM1* (**D**), and *MYBL2* (**G**) in 21 cell types. Correlation analysis of *E2F8* with *MKI67* (**B**) and *RRM2* (**C**), *FOXM1* with MKI67 (**E**) and *RRM2* (**F**), and *MYBL2* with *PLAC8* (**H**) and *IRF8* (**I**).

**Figure 7 ijms-24-13497-f007:**
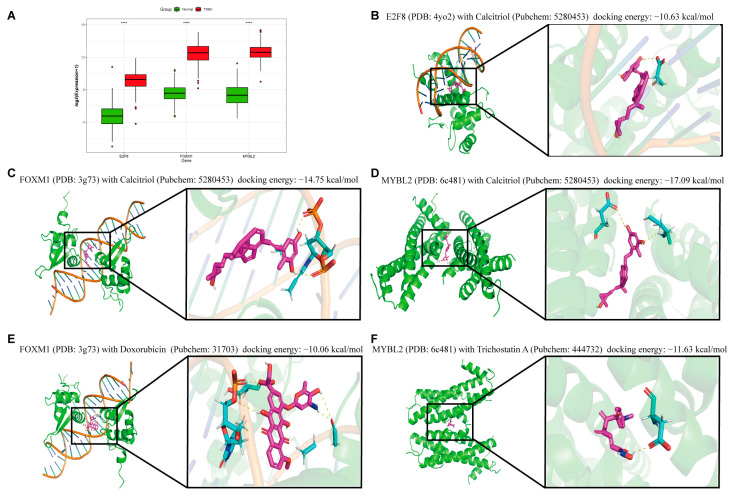
Molecular docking of three Hub TF genes. (**A**) The difference in expression levels of the three Hub TF genes between the TNBC patient and normal groups, *p* < 0.05 was a significant difference (**** *p* < 0.0001). Virtual molecular docking of Calcitriol with *E2F8* (**B**), *FOXM1* (**C**), and *MYBL2* (**D**). Molecular docking of Doxorubicin (DOX) and *FOXM1* (**E**), Trichostatin A (TSA) with *MYBL2* (**F**).

**Figure 8 ijms-24-13497-f008:**
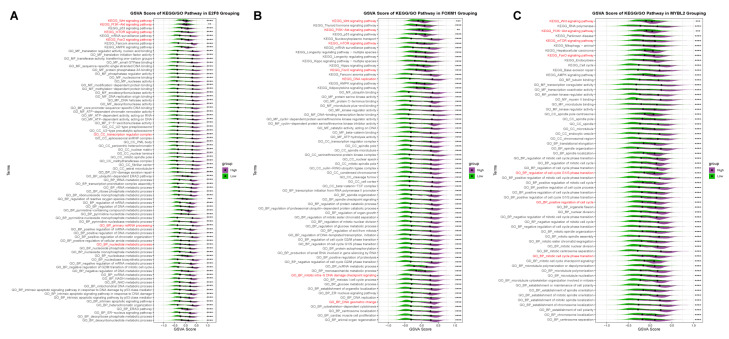
Pathway activity studies of three Hub TF target genes in the TCGA cohort. After *E2F8* (**A**), *FOXM1* (**B**), and *MYBL2* (**C**) were divided into the high-expression group and the low-expression group, the differences in Gene set variation analysis (GSVA) scores of target gene pathways between the two groups were compared. *p* < 0.05 was a significant difference, ** *p* < 0.01; *** *p* < 0.001; **** *p* < 0.0001.

**Figure 9 ijms-24-13497-f009:**
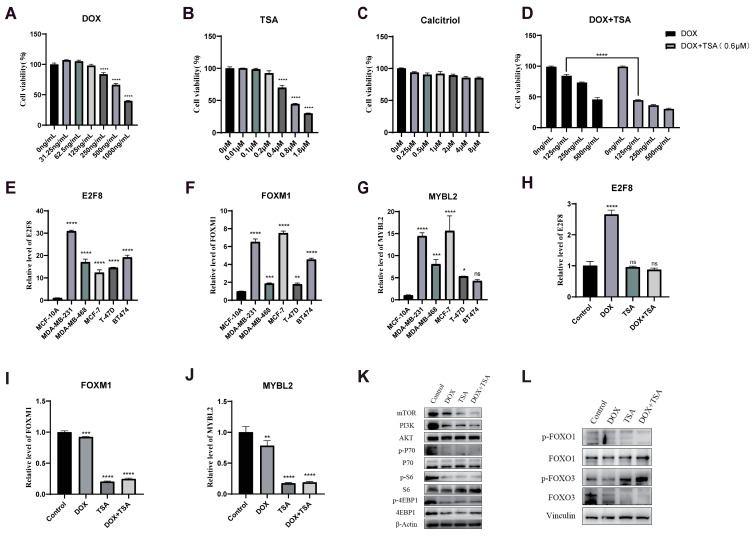
Cell experiments with drug combinations. (**A**) DOX significantly inhibited the proliferation of *MDA-MB-231* cells at different concentrations. (**B**) TSA significantly inhibited the proliferation of *MDA-MB-231* cells at different concentrations. (**C**) The proliferation of *MDA-MB-231* cells was almost unaffected by Calcitriol. (**D**) The combined use of TSA and DOX can significantly reduce the dosage of DOX. TSA = 0.6 μM. DOX = 125 ng/mL, 250 ng/mL, and 500 ng/mL. (**E**–**G**) The expression levels of *E2F8*, *FOXM1*, and *MYBL2* transcription factors in different breast cancer cells. (**H**–**J**) *E2F8*, *FOXM1*, and *MYBL2* transcription factors were expressed in *MDA-MB-231* cells after DOX and TSA treatment. (**K**,**L**) DOX and TSA regulate TNBC progression through PI3K-AKT and FOXO signaling pathways. The *PI3K*, *AKT*, *mTOR*, *p-P70*, *p-S6*, *p-4EBP1*, *p-FoxO1*, and *p-FoxO3* protein levels in *MDA-MB-231* were assessed using Western blotting. Data were shown as means ± SD. The significance of the results was assessed using ANOVA, ns indicates no significance. * *p* < 0.05; ** *p* < 0.01; *** *p* < 0.001; **** *p* < 0.0001 (vs. Control group).

**Figure 10 ijms-24-13497-f010:**
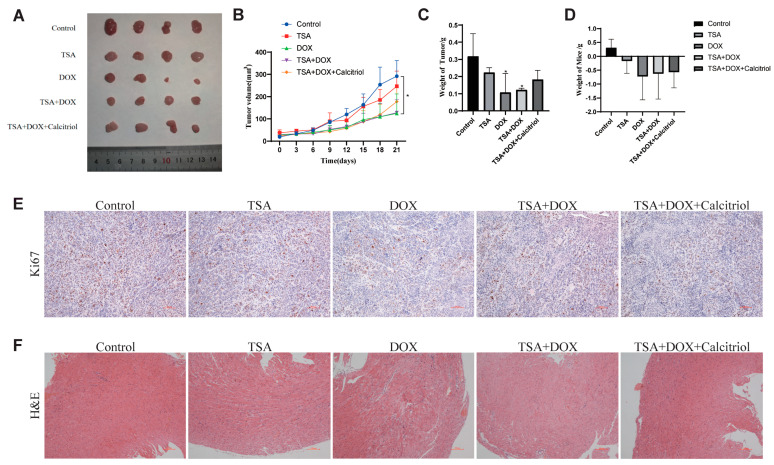
Mouse experiment with the drug combination. (**A**) MDA-MB-231 cells were subcutaneously injected into 6-week-old female BALB/C nude mice. After 21 days of administration, the mice were euthanized and analyzed. Subcutaneous tumors were shown. (**B**) Tumor volume of mice in each group. (**C**) The tumor weight of mice in each group. (**D**) The mice’s weight was measured in each group. (**E**) The expression of Ki67 was detected by IHC staining for Ki67 in the subcutaneous tumor tissue sections. Scale bar, 100 μM. (**F**) Pathological changes in cardiac tissue were observed via HE staining. Scale bar, 100 μM. * *p* < 0.05 (vs. Control group).

## Data Availability

The datasets analyzed in this study are available in GEO and TCGA repositories, including GSE58812, GSE176078, GSE173839, and TCGA-BRCA. The code used in this study is available upon request by contacting the corresponding author.

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
