# Peer review of "Application of Novel Transcription Factor Machine Learning Model and Targeted Drug Combination Therapy Strategy in Triple Negative Breast Cancer"

_ijms, 2023, doi:10.3390/ijms241713497_

Round 1

Reviewer 1 Report

The article entitled "Application of a Novel Transcription Factor Machine Learning Model and Targeted Drug Combination Therapy Strategy in Triple Negative Breast Cancer" by Jianyu Pang et al. is an interesting read. However, there are some issues that need to be addressed.

1. The abstract needs to be rewritten to include the background, aim of the study, materials and methods, results, and conclusion without any lengthy sentences. Additionally, any abbreviations used in the abstract must be defined.

2. The paper lacks organization, and the introduction is too general and lacks specific details about the research gap and significance of the study.

3. Several machine learning (ML) algorithms have been developed to predict the activity of transcription factors (TFs) in cancer cells, including TNBC. It is unclear why the authors chose to use this particular ML model (HTFSS) for their study and how these factors regulate cancer initiation and progression in TNBC.

4. The authors should discuss the prognostic value of HTFSS and its potential clinical implications.

5. While the study used a machine learning model to predict the activity of TFs in cancer cells, the authors did not perform any experiments to investigate the biological mechanisms underlying the activity of the TFs identified in the study. For example, they did not investigate the effects of overexpressing or inhibiting E2F8, FOXM1, or MYBL2 on colony formation and apoptosis, cell cycle changes, and tumor growth in vitro and in vivo.

6. The study showed that HTFSS is a high-risk clinical independent factor. However, the authors did not investigate the potential use of HTFSS as a therapeutic target for TNBC. They did not investigate the effects of inhibiting HTFSS on cell proliferation and tumor growth in vitro and in vivo.

7. The authors need to discuss how the targeted drug combination therapy strategy proposed in this study can be implemented in clinical settings and what the potential ethical implications of using machine learning models to predict drug responses in cancer patients are.

8. The authors should discuss the limitations of this study, and suggestions for addressing them in future research need to be provided. Additionally, the authors should discuss what additional research is needed to validate the findings of this study and further optimize the proposed drug combination therapy strategy.

Extensive editing of English language required

Author Response

Thank you for giving us the opportunity to improve and resubmit our manuscript titled “Application of novel transcription factor machine learning model and targeted drug combination therapy strategy in triple negative breast cancer”. We have carefully studied the comments and suggestions, and then added the corresponding information.  We hope that the revision could be acceptable.

Below is a point-by-point response to the reviewer’comments.

  1. The abstract needs to be rewritten to include the background, aim of the study, materials and methods, results, and conclusion without any lengthy sentences. Additionally, any abbreviations used in the abstract must be defined.

Response 1: The abstract (line 19-39) of this paper has been rewritten in the recommended format, shortening sentences where possible and defining abbreviations when referred to.

  1. The paper lacks organization, and the introduction is too general and lacks specific details about the research gap and significance of the study.

Response 2: Thanks for this suggestion, we rewrote the introduction section (lines 57-71) to pay more attention to organization and to describe the research gap and the significance of the research in specific details.

  1. Several machine learning (ML) algorithms have been developed to predict the activity of transcription factors (TFs) in cancer cells, including TNBC. It is unclear why the authors chose to use this particular ML model (HTFSS) for their study and how these factors regulate cancer initiation and progression in TNBC.

Response 3: As you said, machine learning is widely used to predict transcription factor (TFs) activity. The HTFSS model developed in the present study gives greater attention to the prognostic effect of multiple TFs combined, rather than TF activity per se. HTFSS is also determined by the ratio of four housekeeping genes (ACTB, GAPDH, TFRC, TUBB) and three TFs (E2F8, MYBL2, and FOXM1), which is more applicable than the other machine learning models, i. e. it is not necessary to take into account the quantification of gene expression in the first place. The overarching goal of our study is to advance clinical prognosis through HTFSS while further developing strategies for co-administration of combination targeted drugs to improve current treatment regimens.

The question of how HTFSS factors affect the occurrence and progression of TNBC has been discussed in detail in the discussion section, E2F8: lines 499-505; FOXM1: lines 507-512; MYBL2: lines 515-526.

  1. The authors should discuss the prognostic value of HTFSS and its potential clinical implications.

Response 4: This recommendation is greatly appreciated. In the discussion section, we focus on the prognostic value of HTFSS and its potential clinical significance, which can be seen in lines 454-471.

  1. While the study used a machine learning model to predict the activity of TFs in cancer cells, the authors did not perform any experiments to investigate the biological mechanisms underlying the activity of the TFs identified in the study. For example, they did not investigate the effects of overexpressing or inhibiting E2F8, FOXM1, or MYBL2 on colony formation and apoptosis, cell cycle changes, and tumor growth in vitro and in vivo.

Response 5:  Many thanks for the suggestion. I'm sorry we couldn't catch up on that part of the experiment because of time. Before conducting the drug experiment, however, we conducted a thorough investigation. In previous studies, E2F8, FOXM1, and MYBL2 have been reported to promote cell cycle changes and tumor growth. Since our analysis of pathway activity proved that all three TFs did indeed have promoting effects on cell cycle changes and tumor growth, we performed drug experiments directly. Based on our findings that DOX and TSA effectively inhibited E2F8 expression, FOXM1 and MYBL2 and inhibited the activity of the PI3K-AKT pathway and the FOXO pathway. These results indirectly demonstrated that the up regulation of E2F8, FOXM1, and MYBL2 promoted the activities of these cell proliferation and tumor growth pathways.

  1. The study showed that HTFSS is a high-risk clinical independent factor. However, the authors did not investigate the potential use of HTFSS as a therapeutic target for TNBC. They did not investigate the effects of inhibiting HTFSS on cell proliferation and tumor growth in vitro and in vivo.

Response 6:  Many thanks for your suggestion.  In future studies, it will be important to study in detail the mechanisms by which HTFSS promotes cell proliferation and tumor growth. In this research, we focus on the prognostic value of HTFSS and the feasibility of combining drugs. It is worth mentioning, however, that we targeted several signaling pathways that are significantly affected by the HTFSS: Wnt signaling, PI3K-AKT signaling, FOXO signaling, and mTOR signaling pathways. How the HTFSS acts on these pathways to affect cell proliferation and tumor growth will be explored in greater detail in the future.

  1. The authors need to discuss how the targeted drug combination therapy strategy proposed in this study can be implemented in clinical settings and what the potential ethical implications of using machine learning models to predict drug responses in cancer patients are.

Response 7: This suggestion, which has made great strides in improving our article, is greatly appreciated, because it is more in line with the real application situation, which is what we lacked in the study. A detailed description of how to implement our proposed strategy of targeted drug combination therapy in the clinical setting is provided in the Discussion section of lines 568-575. Lines 485-489 in the discussion section provides an objective answer to ethical questions raised by machine learning. Thanks again!

  1. The authors should discuss the limitations of this study, and suggestions for addressing them in future research need to be provided. Additionally, the authors should discuss what additional research is needed to validate the findings of this study and further optimize the proposed drug combination therapy strategy.

Response 8: Thanks for this constructive suggestion, we have elaborated the limitations of this study, further experimental validation, and optimization of the combination strategy in the last part of the discussion (lines 576-585).

Reviewer 2 Report

This is a very detailed study of the genetic landscape of triple negative breast cancer with extensive detailed results - however it is unclear where the data base used in the study emanates from - the cancer genome atlas is mentioned but significant concerns now exist that the ancestry of the atlas is limited. The introduction section of the paper reads like a methods section and fails to contextualise the paper as a result. The clinical references related to triple negative breast cancer need to be more contemporary and the methods section should follow the introduction. The discussion should lead with a paragraphy summarising the findings , at least one paragraph of the discussion should highlight study limitations

minor edits required

Author Response

Response:

Thank you for giving us the opportunity to improve and resubmit our manuscript titled “Application of novel transcription factor machine learning model and targeted drug combination therapy strategy in triple negative breast cancer”. 

The genome maps involved in this study are from public databases, including TCGA and GEO data, and the specific data index number can be found in 4.1 of the methods section (lines 589-593).

As you said, the ancestry of the atlas is limited, which is an unavoidable problem for all data mining efforts. Therefore, we added validation experiments at the cellular and mouse levels to downstream studies to verify that our cohort study conforms to specific practical results. In the discussion section, the need for the integration of multiple samples (line 576-579) is raised, which will guide future studies to avoid the problem of small sample sizes as much as possible.

Then we have rewritten the introduction to tie context and results together so that it reads more coherently rather than as a single piece.

We have updated some references about TNBC clinical research in recent years, such as Ref. 2, 3, 10, 11, 15. If you have any recommended references, please let us know in time. In addition, the method was placed after the results rather than after the introduction because of the typographical requirements of the journal, which we cannot change and would like you to know.

In the discussion section of lines 454-489, we summarize the results of the study, and in the discussion section of lines 576-585, we give a detailed interpretation of the limitations of this study. Thank you again for your review and proposal of this study.

Reviewer 3 Report

Dear Authors, 

The presented MS ´Application of novel transcription factor machine learning model and targeted drug combination therapy strategy in triple negative breast cancer´.

is very good research work in the field of breast cancer. it contains lots of details and data with a clear presentation. 

some data is not very well visible in figures, so it would be better to use improved captions in figures. not by software generated. 

I liked the discussion and Conclusion section. further details with clinical samples will be very useful in the next study.  

More comments:

The presented MS is focused on the targeted drug delivery system and presents a model system with very high efficiency of detection and sensitivity. However, similar studies and model systems are being developed, author need to clearly specify the advancement in this newly developed system over the previous models.
The presented MS has an edge of finding the gap in the field of drug delivery model, but it needs to add some more discussion in the conclusion to emphasize on the developed model.
Author mentioned in the conclusion that ´ As an independent clinical high-risk factor, HTFSS can be used to predict lymph node involvement in patients with TNBC´ but it is not presented with the following data how?
The author conducted a good search in the references but there is always a room for further addition of some references.
The figures are presented well but it would be a good idea if they find some other approach to present some data of sensitivity and comparison with other methodologies (graphical or table).

Author Response

Response:

Thank you for giving us the opportunity to improve and resubmit our manuscript titled “Application of novel transcription factor machine learning model and targeted drug combination therapy strategy in triple negative breast cancer”.

We have made improvements to the image title, including a more detailed description of the image's meaning, as shown in the Figure 3.

In the introduction (lines 57-71), we describe the differences between our model and other models, and in the discussion (lines 468-471), we highlight the clinical implications and advantages of our model.

Data on "HTFSS as an independent clinical risk factor that can be used to predict lymph node involvement in TNBC patients" are described in Result 2.4 (lines 184-189), specifically to Figure 3C.  Specifically, there are significant differences in HTFSS between different N stages of tumors, which represents the lymph node involvement of patients.  For details, please refer to the TNM Classification of malignant Tumors.

References have been technically checked and updated with a preference for the most recent research reports.

We appreciate this proposal, but due to the lack of time and copyright, we have not been able to find a better way to present our sensitive data.

Round 2

Reviewer 1 Report

Accept in present form

 Minor editing of English language required